# The Mental Health Costs of Armed Conflicts—A Review of Systematic Reviews Conducted on Refugees, Asylum-Seekers and People Living in War Zones

**DOI:** 10.3390/ijerph20042840

**Published:** 2023-02-06

**Authors:** Bernardo Carpiniello

**Affiliations:** Section of Psychiatry, Department of Medical Science and Public Health, University of Cagliari & Psychiatric Unit, University Hospital, 09127 Cagliari, Italy; bcarpini@iol.it

**Keywords:** war, armed conflicts, psychopathology, depressive disorders, anxiety disorders, post-traumatic stress disorders, mental disorders

## Abstract

Aims: Armed conflicts produce a wide series of distressing consequences, including death, all of which impact negatively on the lives of survivors. This paper focuses specifically on the mental health consequences of war on adults and child/adolescent refugees or those living in war zones through a review of all systematic reviews and/or meta-analyses published from 2005 up until the current time. Results: Fifteen systematic reviews and/or meta-analyses conducted in adult populations, and seven relating to children and adolescents, were selected for the purpose of this review. Prevalence rates of anxiety, depression and post-traumatic stress disorder (PTSD) were two- to three-fold higher amongst people exposed to armed conflict compared to those who had not been exposed, with women and children being the most vulnerable to the outcome of armed conflicts. A series of war-related, migratory and post-migratory stressors contribute to short- and long-term mental health issues in the internally displaced, asylum seekers and refugees. Conclusion: It should be a required social responsibility for all psychiatrists and psychiatric associations to commit to raising awareness amongst political decision-makers as to the mental health consequences caused by armed conflicts, as part of their duty of care for people experiencing the consequences of war.

## 1. Introduction

The conflict that ensued following the Russian invasion of Ukraine has served as a stark reminder that war continues to exist, with twenty-seven conflicts currently ongoing worldwide [1]. Of course, death is sadly the most dramatic and acknowledged consequence of war. During the 20th century, armed conflicts were responsible for approximately 191 million deaths [2], mainly civilians. Indeed, the rate of one in seven civilian deaths during World War I rose to two thirds during World War II, with a further increase to 90% of deaths during conflicts at the end of the 20th century [3,4]. Armed conflicts entail a wide series of compelling issues, including negative short- and long- term consequences on mental health, in addition to death, all which impact heavily on the lives of survivors. The onset of conflict-related mental disorders amongst soldiers were first recognized at the beginning of the 20th century [5], although it is generally acknowledged that the most significant mental health impact of war is endured by non-combatant civilians [6]. Taking into account that approximately 2 billion people worldwide currently live in areas involved in armed conflicts resulting in violence, displacement, infrastructural damage and disruption of public health services, a significantly negative effect on their health and wellbeing is to be expected [7]. In addition, available data [8] reveal how 89.3 million people worldwide were forcibly displaced at the end of 2021 as a result of conflicts, persecution, and other forms of violence, with 60% of people remaining in their country as internally displaced persons (IDPs), whilst the remainder crossed international borders as refugees or asylum seekers (According to Amnesty International, internal displacement is the ”involuntary movement of people inside their own country. This movement may be due to a variety of causes, including natural or human-made disasters, armed conflict, or situations of generalized violence”; the asylum seeker is “an individual who is seeking international protection. In countries with individualised procedures, an asylum seeker is someone whose claim has not yet been finally decided on by the country in which he or she has submitted it. Not every asylum seeker will ultimately be recognised as a refugee, but every refugee is initially an asylum seeker”; a refugee is “is a person who has fled their country of origin and is unable or unwilling to return because of a well-founded fear of being persecuted because of their race, religion, nationality, membership of a particular social group or political opinion” (https://www.amnesty.org.au/refugee-and-an-asylum-seeker-difference) accessed on 21 January 2023).

Literature published in relatively recent years clearly depicts the mental health “costs” of conflicts, starting from data relating to the survivors of World War II. Indeed, not counting the millions of deaths and the destruction of entire cities, the Second World War was a highly distressing experience that produced long-lasting consequences on the mental health of survivors, as shown by a study from the World Mental Health Survey Initiative conducted on a sample of 3370 adult civilians living in countries involved in World War II [9]. This study demonstrated how subjects exposed to the conflict had a significantly higher lifetime risk of both major depression (OR = 1.5, 95% Cl 1.1–1.9) and anxiety disorders (OR = 1.5, Cl 1.1–2.0) when compared to those who had not been exposed. Moreover, a higher risk of major depression was detected prevalently in the early post-war years, while the risk of anxiety disorders increased over time. The authors of this study concluded that their data demonstrate an increased risk for common mental disorders in people exposed to the Second World War and hypothesize as to whether comparable patterns might be observed amongst civilians affected by more recent conflicts.

Starting from these premises, the present paper aims to focus specifically on the mental health consequences of war on adults and child/adolescent refugees or those still living in war zones through a review of systematic reviews and/or meta-analyses published in English from 2005 up until the current time.

## 2. Methods

### 2.1. Eligibility Criteria

For the purpose of this review, we considered only systematic reviews and/or meta-analyses published in English from 2005 to 2022, reporting quantitative data about the prevalence of mental disorders in adults and children living in conflict areas, or refugees/asylum seekers exposed to war and/or armed conflicts.

### 2.2. Sources of Information and Search Strategy

An electronic search of the literature was performed by the author using PubMed and Scopus as data bases. The key words used were “armed conflicts” and “mental health”, “armed conflict” and “mental disorders”, “war” and “mental health”, “war” and “mental disorders”. Eighty-three and 57 papers were retrieved, respectively, in Scopus and PubMed, using “armed conflicts” and “mental health” as key words; using “armed conflict” and “mental disorders” we retrieved respectively 56 and 51 papers; using “war” and “mental health” we found respectively 50 and 104 papers; and using “war” and “mental disorders” as key words, 488 and 114 contributions, respectively, were identified.

### 2.3. Selection and Data Collection Process

Titles and abstracts were screened by the author for inclusion. Articles rated as possible candidates were added to a preliminary list and their full texts were retrieved and examined. A total of 22 papers were finally selected and considered for this review, following elimination of duplicates and papers not included in the category of systematic reviews and/or meta-analyses. Fifteen papers are related to adult populations only, two papers regard both adults and child/adolescents, and five are related to children and adolescent populations.

## 3. Results

### 3.1. Systematic Reviews and/or Metanalyses Relating to Adult Populations

Seventeen systematic reviews and/or meta-analyses on adult populations have been published between 2005 and the current time. The main characteristics of these studies are reported in Table 1. Prevalence rates of mental disorders in systematic reviews/meta-analyses concerning adult refugees/asylum seekers are reported in Table 2, while the prevalence rates of mental disorders in systematic reviews/meta-analyses relating to samples of refugees living in war/conflict-affected countries are reported in Table 3. All systematic reviews/metanalyses taken into consideration include almost exclusively cross-sectional studies, thus reporting mostly point or period prevalence rates. Considering specifically each single study, the systematic review by Fazel et al. [10], considers only large studies (at least 200 subjects examined through standardized interviews), reporting a prevalence rate of approx. 9% (99% CI 8–10%) for post-traumatic stress disorder (PTSD), 5% (CI 4–6%) for major depression (MD) and approximately 4% (CI 3–6%) for generalized anxiety disorders (GAD) amongst adult refugees. From a total of four studies reporting data relating to comorbidity, 71% of subjects diagnosed with MD had also received a diagnosis of PTSD, and 44% of those diagnosed with PTSD were also affected by MD. The prevalence rate of psychotic disorders, based on data from only two studies, was approximately 2% (CI 1–6%), based on data from only five surveys, PTSD prevalence among children and adolescents was 11% (CI 7–17%). According to the authors, approximately one in ten adult refugees in western countries is affected by PTSD, and approximately one in 20 has MD, and one in 25 has a GAD, with a possible overlapping of these disorders in many people.

A meta-analysis by Porter and Hasmal [11] reports the effect size estimates for a comparison between refugees and non-refugees, revealing a weighted mean effect size of 0.41 (range, −1.36 to 2.91), indicating that refugees had moderately poorer psychopathological outcomes. Interestingly, this difference remained, even when the sample considered was constituted by people who had been displaced or directly exposed to war and violence, thus confirming the depth of distress that refugees experience. Moreover, the study also highlights how a series of pre-displacement characteristics in refugees (older age, better education, female, with higher socioeconomic status and rural residence), as well as several post-displacement characteristics (living in institutional accommodations, having restricted economic resources, internal displacement within their own country, or being repatriated to their country in the presence of ongoing conflict) were related to worse outcomes. According to the authors, these results highlight how contextual factors, both before and after displacement, may be considered as moderators of refugees’ mental health. Thus, mental health consequences produced by displacement during armed conflicts should not be considered solely as the product of an acute and significant stressor, particularly given the relevant role of the socio-economic and cultural contexts from which refugees are displaced and to which they are subsequently reassigned.

A study by Steel et al. [12] focuses in particular on PTSD and depression. The rates of these disorders in this study display a wide inter-survey variability, mainly due to methodological factors, with non-random sampling, small sample sizes, and self-report questionnaires associated with higher rates of mental disorder. The reported unadjusted weighted prevalence rate for PTSD was 30.6% (95% CI, 26.3–35.2%) and 30.8% (95% CI, 26.3–35.6%) for depression. After adjustment for methodological factors, reported torture emerged as the strongest factor associated with PTSD (OR, 2.01; 95% CI, 1.52–2.65), followed by cumulative exposure to other potentially traumatic events (PTEs), (OR, 1.52; 95% CI, 1.21–1.91), time since conflict (OR, 0.77; 95% CI, 0.66–0.91), and assessed level of political terror (OR, 1.60; 95% CI, 1.03–2.50). With regard to depression, significant factors included the number of PTEs (OR, 1.64; 95% CI, 1.39–1.93), time since conflict (OR, 0.80; 95% CI, 0.69–0.93), reported torture (OR, 1.48; 95% CI, 1.07–2.04), and residency status (OR, 1.30; 95% CI, 1.07–1.57). Thus, the findings of this study clearly illustrate how exposure to torture and other PTEs may be considered major risk factors for PTSD and depression among refugees and conflict-affected populations. These results are of particular interest, even in light of the data from a systematic review showing reported prevalence rates of torture ranging from 1% to 76% (median 27%) among “forced” adult migrants living in high-income countries [13]. 

Bogic et al. [14] investigated the long-term consequences of war, including in their systematic review, and only war-refugees examined at least five years after displacement from their country of origin. The authors found current prevalence rates of an unspecified anxiety disorder ranging from approximately 2% to 88%, depression between 2.5% and 80%, and PTSD between 4.5% and 86%. Excluding methodological reasons, the wide variability of data was related to the country of origin and of destination: pre-migratory, war-related and post-migratory stress constituted the most relevant risk flactors for psychopathology. 

A systematic review by Ba and Bhopal [15] places a specific focus on the physical, mental, and social consequences of war-related sexual violence (individual rape, gang rape, sexual slavery) among civilians living in countries involved in armed conflicts, mainly in Africa. The majority of studies were carried out on females, although four studies included males. Data on mental disorders were reported in 14 out of 20 studies. The detected range of prevalence rates were 3.1–75.9% for PTSD, 6.9–75% for anxiety disorders and 8.8–76.5% for depression. Again, the high variability of reported prevalence rate, according to the same authors, may be due to a series of methodological factors such as the different characteristics of the samples in terms of age, gender, country of origin, difference in research tools and the recall period, and to non-methodological factors, such as the different types of sexual violence suffered by victims and the eventual multiplicity of traumatic experiences. Incidentally, this study likewise reports high rates of physical consequences of violence such as pregnancies, genital injuries, sexually transmitted infections and sexual dysfunctions. Rejection by family or community (3.5–28.5%) and spousal abandonment (61.0–64.7%) were amongst the negative social outcomes reported.

The systematic review by Morina et al. [16] deals with studies conducted to investigate common mental disorders amongst adult refugees and “internally displaced people”, i.e., people who had moved from their area of residence to another place in the same country or geographical area to escape conflict. A wide variability of prevalence rates was reported for PTSD (3–88%), depression (5–80%), and anxiety disorders (1–81%). Twelve original articles out of the 38 initially considered reported data relating to other mental disorders. In particular, the prevalence rates for alcohol use disorder and drug use disorders of 2.0–65.0% and 2.0–20.0%, respectively, were found. Prevalence rates for psychoses was 1–12% (data from only two studies), while “psychotic symptoms” and “hallucinations” accounted for 13% and 21%, respectively (data from only one study). Reported prevalence of suicidality ranged from 2.0 to 12.0% (data from only one study). Rates of pain and somatoform disorders ranged from 14.0% to 29.0% (data from two studies).

A meta-analysis by Morina et al. [17] conducted solely on studies which had adopted structured clinical interviews, reported a prevalence of both depression and PTSD in civilian survivors living in a war zone. The pooled prevalence rates obtained for depression and PTSD were 27% and 26%, respectively, with 10% of subjects affected by both disorders. A higher prevalence of depression was found in surveys comprising participants with a higher mean age, and of PTSD in surveys featuring higher rates of unemployment and higher percentages of women; on the contrary, a lower prevalence of PTSD was found in samples with a higher number of participants living with a partner. According to the authors, even when taking into account a series of limitations of the studies considered, both depression and PTSD were found to be highly prevalent in survivors living in war zones. 

The meta-analytic study carried out by Charlsson et al. [18] is one of the few to report data highlighting the prevalence of common mental disorders in relation to their severity and functional impact. The authors report considerably high age-standardised prevalence rates for depression (10.8% [95% UI 8.1–14.2], PTSD (15.3% [9.9–23.5]), and anxiety disorders (21.7% [16.7–28.3]) among conflict-affected populations, with mild forms of these disorders being the most widely prevalent. Indeed, mean comorbidity-adjusted, an age-standardised point prevalence of 13.0% (95% UI 10.3–16.2) was reported for depression, anxiety, and post-traumatic stress disorder as a group, versus 4.0% (95% UI 2.9–5.5) for moderate forms and 5.1% (95% UI 4.0–6.5) for severe forms. Moreover, the authors estimated that at any time point, approximately 9% of conflict-involved populations were affected by a moderate to severe mental disorder. Prevalence trends for depression and anxiety increased with age, as did prevalence rates for depression, post-traumatic stress disorder, and any anxiety disorder in women. Moreover, this study also reported a series of data relating to functioning: the estimated age-standardised years lived in disability (YLDs) for depression in conflict-affected populations was 24.8 YLDs per 1000 population (95% UI 16.4–36.0), versus the global age-standardised estimate of 4.6 YLDs per 1000 population (3.2–6.2) reported by the 2016 Global Burden Disease (GBD) study; the estimates of YLDs for any anxiety disorder were 23.2 YLDs per 1000 population (95% UI 17.0–29.9), versus the GBD 2016 estimates of 3.5 YLDs per 1000 population (2.5–4.8).

Hoppen and Morina [19] were the first to attempt to estimate in absolute numbers how many adult war survivors globally might be affected by PTSD and/or MD. Based on the prevalence rates for these disorders obtained by their meta-analysis of randomized epidemiological surveys on survivors living in war zones between 1989 and 2015, the authors extrapolated the absolute numbers of those affected by mental disorders, using general population data from the United Nations. Thus, they estimated that approximately 1.45 billion individuals worldwide had experienced war throughout the 25 years and were still alive in 2015, including one billion adults; moreover, calculating that approximately 354 million adult war survivors were affected by PTSD or MD, with approximately 117 million suffering from comorbid PTSD and MD.

The meta-analytic study by Blackmore et al. [20] adopted strict inclusion criteria for the diagnosis of mental illness in current refugee and asylum-seeker populations, including only studies reporting the prevalence of mental illness based on clinical interviews with trained assessors, and using validated diagnostic measures. Moreover, the study is the first systematic review devoid of restrictions on language, countries of origin, or settlement. The study focuses not only on PTSD, but also on depression, anxiety, and psychosis. The prevalence rates reported in this study were 31.46% (95% CI 24.43–38.5) for PTSD, 31.5% (95% CI 22.64–40.38), for depression, 11% (95% CI 6.75–15.43) for anxiety disorders and 1.51% (95% CI 0.63–2.40) for psychosis. Subgroup analyses were performed, showing a significant difference according to sex, sample size, displacement duration, visa status, country of origin, current residence, type of interview (interpreter-assisted or native language), and diagnostic measure. Of relevance, these analyses showed a higher PTSD prevalence for women, a higher prevalence of PTSD and depression in studies with interpreter-assisted diagnostic assessments, and a more pronounced persistence of PTSD and depression for many years post-displacement. Overall, the study showed that refugees and asylum seekers experience higher rates of mental illness, in particular PTSD and depression, compared to those found in the general population. 

The meta-analytic study conducted by Ng et al. [21] focused specifically on PTSD in samples of people living in 10 of the 48 countries of the Sub-Saharan area, a region of Africa considered to be disproportionately affected by individual and population-level exposure to trauma. According to this meta-analysis, the pooled prevalence of probable PTSD across all studies was 22% (95% CI 13–32%). The sub-analysis was performed based on an exposure or lack of exposure to war and reported an estimated PTSD pooled prevalence of 30% (95% CI 20–40%) among war-exposed populations and only 8% (95% CI 3–15%; *p* = 0.01) for those living in regions not exposed to war. Based on these results, a traumatic wartime experience clearly represents the main risk factor for mental health disorders in these populations.

A study performed by Henkelmann et al. [22] reports prevalence rates of PTSD, anxiety (AD), and depressive disorders (DD) amongst refugees living in high income countries according to the methods of evaluation used in the studies (clinical interviews or self-reported instruments). The average prevalence rates were 30% for AD, 36 for DD and 34% for PTSD. As expected, prevalence rates were respectively 13 and 42% (95% CI 8–52%) for diagnosed versus self-reported AD, 30% vs. 40% (95% CI 23–48%) for diagnosed versus self-reported DD, and 29% vs. 37% (95% CI 22–45%) for diagnosed versus self-reported PTSD. Prevalence rates of AD, DD and PTSD did not differ, in line with a series of variables including continent of origin or resettlement, average duration of residence, mean age and gender distribution of the sample, and methodological quality of the study. The authors reported finding high rates of prevalence of AD, DD and PTSD amongst adult refugees, not only compared to non-refugee populations, but also to populations living in conflict or war zones. This seems to suggest that exposure to conflict and war is not the only factor to increase refugee vulnerability, particularly as flight and/or additional post-migration factors may aggravate trauma-related symptoms.

The study carried out by Hoppen et al. [23] to assess the prevalence of PTSD and major depression (MD) in representative samples of people living in countries with a recent history of war (1989–2019) is of particular interest, due to the extrapolation of the absolute global numbers of affected people and disability-adjusted life years associated to the disorders (DALYs), drawing on the data of the Global Burden of Diseases Study 2019. According to this meta-analytic study, the PTSD point prevalence was 26.51% and 23.31% for MD. Of those affected by PTSD, 55.26% presented with comorbid MD. Prevalence rates were not significantly associated with intensity and length of war, time since war, response rate, or survey quality. The extrapolation of prevalence data yielded a total of 316 million adult war-survivors globally affected by PTSD and/or MD in 2019, almost all of whom resided in low/middle-income countries. The burden of disease due to PTSD and MD were 3,105,387 and 4,083,950 DALYs, respectively.

The systematic review and meta-analysis performed by Stein et al. [24] is focused solely on studies investigating post-traumatic stress (PTS) in adult civilians living in the Eastern Mediterranean region, considered the area of the world’s most violent conflicts and the origin of the largest refugee population worldwide. The pooled estimate of PTSD syndromes was 31% (95% CI: 27–35%), with a considerable heterogeneity and a huge true rate interval (1–76%). This heterogeneity in the rates of PTSD was attributed by the authors to a series of factors, analytically evaluated, related to sample characteristics (i.e., sample size, country of origin), methodological (i.e., decade of study, type of assessment), conflict-related (i.e., stage and severity) and displacement-related factors (present/not present; site of displacement: western or non-western countries).

Scoglio and Sahli [25] undertook a systematic review of mental health problems in refugees who had resettled in high-income countries after previous exposure to violence. The authors reported how these populations were much more likely to display higher rates of exposure to violence compared to control groups, and how the mental health consequences produced, in terms of symptom classes, comprised prevalently of anxiety, depressive or post-traumatic stress disorders, and with an overall prevalence of “mental health problems” ranging from 55% to 75% of samples considered in the studies.

The systematic review and meta-analysis by Mesa-Vieira et al. [26] focused specifically on migrants exposed to pre-migratory armed conflict who were assessed using standardized interviews. This study estimated a current prevalence of 31% (95% CI 23–40) for PTSD, 25% (95% CI 17–34) for MD, and 14% (95% CI 5–35) for AD. Age at migration, income of the country of origin and host country, intensity of conflict, and time since displacement were all variables associated with an increased prevalence of mental health disorders.

Finally, the meta-analysis of Lim et al. [27] reports data on prevalence rates of depressive, anxiety and post-traumatic stress disorders among people living in war and conflict-affected areas, including military populations and general populations (civilians or refugees). This study is the only one that reports not only the ranges and the aggregate prevalence rates of the disorders considered, but also a series of specific analyses regarding civilian vs. military subgroups and studies relating to war periods vs. post-war periods. This study shows a large variability of prevalence for all disorders, ranging from 3.2 to 79.6% for depression, from 4.2 to 94.8% for anxiety, and from 3.9 to 69% for PTSD, but also that the aggregate prevalence rates, according a random-effects model, are higher for depression and anxiety (respectively 28.9% and 30.7%) than for PTSD (23.5%). Interestingly, rates are very significantly higher (*p* < 0.001) among civilians than military populations in regard to depression and anxiety (respectively 33% vs. 24% and 38.6% vs. 16.2% for anxiety); with a non-significant difference for PTSD (respectively 25.7% vs. 21.3%). Of outmost importance are the differences of prevalence during war and post-war periods, which are univocally higher (*p* < 001) during war (respectively 38.7% vs. 29.1% for depression, 43.4 vs. 30.3% for anxiety).

### 3.2. Systematic Reviews and/or Meta-Analyses Relating to Children/Adolescents

A total of seven systematic reviews and/or meta-analyses have been published from 2005 up until the current time, as reported in Table 4. The prevalence rates of mental disorders emerging from these studies are summarized in Table 5.

The previously cited systematic review by Fazel et al. [10] was based specifically on five studies carried out on 260 children or adolescent refugees in Canada, Sweden, and the USA, who had originated from Bosnia, Central America, Iran, Kurdistan, and Rwanda. According to this review, 11% (7–17%) of refugee children were affected by post-traumatic stress disorder. No study reported data on depression. A more detailed analysis of data obtained for adolescents and young adults revealed prevalence rates of PTSD and MD in this group of 35% and 12%, respectively; therefore, higher, although not significantly, than those found in adult populations (respectively, 10% for PTSD and 6% for MD).

The systematic review and meta-regression study performed by Attanayake et al. [28] reported a wide prevalence range for PTSD (from 4.5% to 89.3%), with a pooled estimate of 47% (9% CI: 35–60%, I2 = 98%), with this heterogeneity being attributable to study location, method of measurement and duration since exposure to war. The pooled rate of depression, based on four studies alone was 43% (95% CI: 31–55%), and rate of anxiety disorders 27% (95% CI: 21–33%), based on three studies. The authors concluded by underlining a higher prevalence rate of mental disorders among children exposed to conflict compared to rates found in the general population.

The systematic review carried out by Dimitry [29] focused on the mental health of children and adolescents living in areas of armed conflict in the Middle East (Israel, Palestine, Lebanon and Iraq), although data from Lebanon was very limited. The study reported how children and adolescents living in these conflict zones had been exposed to high levels of traumatic experiences, showing in particular how the prevalence of mental, behavioural and emotional problems appears to correlate positively with the amount of war-related traumatic experiences. The review reported data for each single area considered in the study. With regard to Israel, the review reported a prevalence of post-traumatic stress disorder ranging between 5% to 8% and of mild depression between 25% to 35% (with 3.3% of adolescents affected by major depression). The authors also reported a prevalence of 3%, 2.5%, 1.8%, 1.4% and 1.2% for attention-deficit hyperactivity, specific phobias, oppositional defiant disorder, generalized anxiety disorder and obsessive-compulsive disorder, respectively. Moreover, separation anxiety in children was reported in more than 80% of the samples, with a very high prevalence (more than 50–60% of cases) of other non-specific problems such as nervousness, agitation or aggressiveness, excessive crying, nightmares or anxious arousal. Similarly, considerably high rates of externalizing problems such as risk-taking behaviours were reported for adolescents. The study reported for Palestine reported a prevalence rate for PTSD of 23–70%, with mild cases ranging from 7% to 48% and moderate–severe cases between 39% to 89%; high anxiety levels were reported in 40–100% of samples, while the reported prevalence of depression, conduct disorder and ADHD were, respectively, 11.3%; 14%, and 10%. The presence of “emotional disorders” was reported in 47% of samples, with the prevalence of one or more disorders in 51% of cases. Finally, 28% of Palestinian children displayed a fear of leaving the house. With regard to Iraq, PTSD prevalence rates of 10–30% were reported, with separation anxiety disorder, conduct disorders, specific phobias rates of 4.3%, 2.5%, and 3.3%, respectively. The estimated overall prevalence of mental disorders in this country ranged from 36% to 38%. The authors identified the main determinants of psychopathology as level and type of exposure, age, gender, socio-economic adversity, levels of social support, and religiosity.

The systematic review by Slone and Mann [30] was based mainly on studies reporting prevalence data relating to behavioural issues such as disturbed play or disturbed parent-child relations, and emotional symptoms (including psychosomatic symptoms) rather than focusing on specific syndromes. Moreover, in addition to children exposed to war or armed conflicts, this review also took into account children exposed to acts of terrorism, such as the twin towers attack.

The above-cited systematic review and meta-analysis by Henkelman et al. [22] included supplementary material, providing specific data relating to the prevalence of anxiety disorders (five studies, 493 subjects), depression (seven studies, 995 subjects) and PTSD (7 studies, 662 subjects) detected in child and adolescent samples. Self-reporting instruments were used in the majority of studies (four out of five for anxiety, six out of seven for depression, and six out of seven for PTSD). The average prevalence rates were 32% (28–37%) for anxiety (versus 28% for adults), 28% (19–37%) for depression (versus 36% for adults) and 52% (35–68%) for PTSD. Prevalence rates estimated for adults using self-reporting instruments were generally higher than those obtained through clinical diagnosis, although a reverse pattern was reported for child/adolescent samples. Indeed, self-reported vs. clinically diagnosed prevalence rates of 31% vs. 50% were reported for anxiety, 27% vs. 335 depression and 38% vs. 80% for PTSD. Interestingly, the study also reported prevalence data for samples made up of refugees (Re) and children/adolescents living in conflict areas (Ca). Accordingly, rates of reported anxiety corresponded to 32% (Re) and 27% (Ca), respectively rates of depression were 28% (Re) and 43% (Ca); and rates of PTSD were 52% (Re) and 47% (Ca). The authors of the study concluded that their estimated prevalence rates were substantially higher in adult and child/adolescent refugees compared to those reported for non-refugee populations worldwide and populations living in conflict or war zones.

The systematic review by Kien et al. [31] comprised studies focussing on asylum-seeking children and adolescents, or refugee minors, conducted in European countries. The authors reported the results narratively, as meta-analyses were impeded due to high heterogeneity of the studies. Point prevalence estimates were reported as medians and interquartile ranges. Prevalence estimates were 15% (8.7–31.6%) for anxiety disorders, 20.7% (19.3–32.8%) for depression, and 35.3% for PTSD (19.0–52.7%). The authors also reported a 25.2% (19.8–35.0%) prevalence rate for “emotional and behavioural problems”, and a rate of “suicidal ideation and behaviours” of 5.0% (0.7–9.3%). The authors commented how, compared to the general population, the prevalence of psychiatric disorders and mental health problems found in their study was substantially higher, noting that participants had generally originated from countries affected by war (e.g., African states such as Somalia, Afghanistan, and Balkan states), which could explain the particularly high estimates for PTSD.

Finally, a systematic review and meta-analysis by Blackmore et al. [32] included studies undertaken in five countries (Germany, Malaysia, Norway, Sweden and Turkey) on refugee samples hailing from the Middle East, African countries, and Southern Asia. Studies selected only included those adopting clinical interviews for the purpose of diagnosis. The overall prevalence rates estimated in this study were 22.7% (95% CI 12.79−32.64) for PTSD, 13.8% (95% CI 5.96−21.67) for depression, 15.8% (95% CI 8.04−23.50) for anxiety disorders, 8.6% (1.08−16.12) for attention-deficit/hyperactivity disorder (ADHD) and 1.69% (95% CI −0.78 to 4.16) for oppositional defiant disorder (ODD). A higher prevalence of mental disorders was obtained for refugee and asylum-seeker populations compared to non-refugee populations, particularly in subjects who had been displaced for less than 2 years.

## 4. Discussion

Prior to discussing the data arising from our review, the extensive heterogeneity of individual studies on which all systematic reviews and meta-analyses are based should be highlighted. The frequently large variability of results may be due in part to a series of methodological and clinical factors. Methodological factors may include sources of variability such as sample size and sampling strategies (i.e., random sampling vs. non-random sampling), and evaluation methods (i.e., use of structured clinical interviews vs. self-reporting questionnaires; dimensional vs. categorical approaches to diagnoses; use, or lack of culture-adapted questionnaires; and clinical interviews administered in the original language of the samples examined or using a trained interpreter). However, even taking into account this large variability of data, our review of systematic reviews/meta-analyses clearly yielded a significantly higher prevalence rate of current mental disorders for the adult and younger populations who are still living in conflict areas or have been displaced as “forced migrants” (refugees or asylum seekers), compared to the populations who have not been exposed to war, as also shown by the umbrella review from Turrini et al. [33] who found point estimates ranging from 4% and 40% for anxiety, from 5% to 44% for depression and from 9% to 36% for PTSD, affecting, on the average, approximately one third of these populations. Overall, these data confirm the heavy emotional distress among those who leave their homes, culture, and communities due to threats of violence and persecution [34]. Prevalence of depression and anxiety seems to be even higher than PTSD, and more represented among civilians than to military personnel; moreover, rates of common mental disorders seem to gradually decrease in post-war periods [27].

In more concrete terms, millions of people exposed to war-related stressors are affected by mental disorders [19], thus producing a heavy burden of psychopathology in terms of disability [18,23]. Prevalence of depression and anxiety seems to be even higher than PTSD, and more represented among civilians than to military personnel; moreover, rates of common mental disorders seem to gradually decrease in post-war periods [27]. The heavy toll paid by civilians living in war zones in terms of psychopathology, in particular anxiety disorders, depressive disorders and post-traumatic stress disorders, is unequivocal [27,35,36] and may be attributed both to direct exposure to the distress of war and to a series of displacement-related stressors [37], as depicted in Table 6. Undoubtedly, pre-migration traumatic experiences such as those directly linked to war and conflicts are important predictors of negative mental health outcomes, but a series of post-migration stressors play a very relevant role for refugees and asylum seekers. Among these socioeconomic factors (i.e., unemployment or underemployment, financial restrictions/poverty, lack of secure housing), social and interpersonal factors (i.e., family separation, change in previous social role, social isolation, discrimination, loss of social identity, lacking social support, changes in gender role), factors related to the asylum process and to immigration policies (i.e., mandatory detentions, extended processing times, insecure visa status, lack of access to legal services and representation) have been considered of outmost importance [38].

Although the numerous surveys included in the systematic reviews/meta-analyses largely report on current prevalence, the persistence of war-related psychopathology is confirmed by the study conducted by Bogic and al. [14] on refugees assessed at least five years following resettlement, as well as by other studies of samples of people who had lived through the Second World War during their youth and were assessed in adulthood or late adulthood [9,39,40,41,42]. The consequences produced by war on the survival and physical and mental health of exposed populations assumes a dramatic profile amongst the ageing and female populations [43,44]. This review revealed the particular vulnerability of women and children to develop conflict-associated mental health sequelae correlated with both the degree of trauma, and availability of physical and emotional support [43]. In particular, in women, the higher risk of war-related mental health consequences seems to be largely related to sexual and physical violence [15,43]. The severe consequences of armed conflicts for minors emerging from our review should be interpreted the light of the large variability of individual experiences between different subjects [45] and of several specific factors, including the fact that prior to displacement, children and adolescent survivors may have been exposed not only to armed conflicts and threats, but also to separation from family members and the death of parents and close relatives [46]. It should also be taken into account that many young people may also have been required to fight as “child soldiers” [47,48], thus adding to the burden of traumatic experiences. Family violence, moreover, may represent a further source of war-related distress commonly suffered by children and women, taking into account that the prevalence of violence against women in the context of conflict tended to be in the range of 30–40%, with more than seventy-five percent of children having experienced violence [49]. The role of parenting in war zones has been investigated in relation to mental health issues manifested in their offspring, given that parents exposed to war may show less warmth towards their children and treat them more harshly, thus partly mediating the association between exposure to war and child adjustment, in terms of traumatic stress symptoms, depression and anxiety, social problems, and externalizing behaviour [50]. To summarize, mental health issues manifested in minors as the consequence of armed conflicts may represent part of a complex process correlated with the stage of exposure, length of conflict, and other contextual factors [51]. Lastly, in line with the findings of a study of 1966 adults whose fathers had served in the Australian army during the Vietnam War, the possibility of a transgenerational transmission of the consequences of war on mental health should be taken into consideration, particularly as this study demonstrated, almost 40 years after the war, how the adult offspring of deployed veterans were more likely to be diagnosed with anxiety and depression, displaying suicide ideation, suicidal plans and self-harm behaviours more frequently than the progeny of comparable, non-deployed army veterans [52].

## 5. Conclusions

A series of limitations in the current literature should be acknowledged, such as the often-low quality of studies and the paucity of longitudinal studies, with the consequent difficulty of elucidating the directionality of the relationship between postmigration stressors and mental health consequences, and the relative paucity of studies regarding wider ranges of outcomes beyond PTSD, depression and anxiety [33,38]. Nevertheless, evidence from the literature unequivocally demonstrates that war and the consequent displacement are among the most challenging threats to mental health, particularly for the innocent civilian victims of conflicts. Therefore, early mental health care for people suffering from war-related mental disorders should be considered a priority [53], in particular, with regard to children and adolescents [38,54,55]. Unfortunately, the encouraging evidence in favour of the efficacy and acceptability of psychosocial interventions in asylum seekers and refugees is limited to adult populations [56]. Moreover, in spite of the availability of effective treatments, on practical grounds, the low healthcare access and the poor healthcare services available for vulnerable populations such as migrants and refugees [56] are the major obstacles for protecting mental health of these populations. In particular, a series of barriers to mental health care access have been described [34,57,58], such as the lack of knowledge of legal entitlements and the health care system in the hosting country, the poor command of the hosting country language, the negative cultural beliefs about mental health, the different cultural expectations towards health care professionals, the lack of trust towards services and authorities in the hosting country, and finally, the fear of stigma. In order to overcome these barriers and to improve mental health care, a series of recommendations have been suggested [55], namely the ensuring of education about self-recognition of symptoms of mental health disorders; the overcoming of the social and linguistic barriers to access mental health services; the provision of services for the wellbeing, augmenting safe, socially acceptable basic services with the help of local support groups and non-governmental agencies; the strengthening of the screening capacity for mental health disorders at emergency and primary care level; the provision of specialized services by mental health professionals; and the increase of telepsychiatry utilization. Moreover, considering the deleterious impact of post-migration difficulties and the consequent need for psychosocial interventions not limited to a trauma-focused perspective, Li et al. [38] suggested policy developments in order to mitigate detrimental post-migration factors through the modification of migration and social policies in hosting countries. Examples of these modifications are policies and programs to support refugees to find employment, to build social capital and connections in their new communities, the setting of limitations to prolonged detention in camps and to the ongoing temporary protection. On the same grounds, Evangelidou et al. [59], in the context of the MyHealth Project, recommend, at the level of local authorities of destination countries, the facilitation of administrative issues (i.e., creation of work or educational opportunities), the promotion of community-based activities of social integration (i.e., using “expert peers”, establishing health/social services for vulnerable refugees, making available preventive health interventions such as peer support groups, raising awareness in local neighbourhoods about immigration issues, discrimination, racism) and investing in healthcare professional skills (i.e., interpreters, cultural mediators). At the level of target populations (migrant/refugees), the actions recommended are those devoted to language acquisition, cultural adaptation of help-seeking behaviours patterns, or proactiveness in networking with local communities. Finally, psychiatrists and psychiatric associations in general are duty-bound to safeguard the mental health of the population [60], not only by taking care of those in whom mental health is impacted by war and post-war-related consequences, but also as a social responsibility to be undertaken through a commitment to raising awareness amongst political decision-makers as to the mental health consequences of armed conflicts and to the mental health policies to be adopted.

## Figures and Tables

**Table 1 ijerph-20-02840-t001:** Systematic reviews and/or meta-analyses relating to the mental health consequences of conflicts conducted on adult samples.

Authors, Year of Publication (Reference Number)	Study Type	Number and Time Frame of Papers Considered	Psychopathological Conditions Considered	Study Samples (N, Type of Populations Considered)
Fazel M, et al., 2005 [10]	Systematic review	20/1996–2002	prevalence of PTSD, major depression, generalised anxiety and psychotic disorders	N = 6743 refugees resettled in high-income western countries
Porter M and Haslam N, 2005 [11]	Meta-analysis	56/1959–2002	Comparison of measures of psychopathology among refugees versus non-refugees	N = 22,221 refugees and N = 45,074 non refugees
Steel Z et al., 2009 [12]	Systematic review and meta-analysis	161/1980–2009	Prevalence rates of PTSD and depression and associated factors	N = 81,866 refugees and other conflict-affected persons
Bogic M et al., 2015 [14]	Systematic review	29/1993–2013	Prevalence rates of anxiety disorders, depressive disorders, PTSD	16,010 war-affected refugees evaluated 5 years or more after displacement
Ba I and Bhopal RS, 2017 [15]	Systematic review	20/1981–2014	Prevalence rates of mental disorders	N = 42,289 civilians exposed to war-related sexual violence
Morina N et al., 2018 [16]	Systematic review	38/1996–2016	Prevalence rates of common mental disorders	N = 39,518 internally displaced and refugee adults from 21 countries
Morina N et al., 2018 [17]	Systematic review and meta-analysis	33/1994–2015	Prevalence rates of PTSD and depression	N = 24,896 survivors who had lived in a war zone up to 25 years prior to the time of the survey
Charlson F et al., 2018 [18]	Systematic review and meta-analysis	129/2000–2017	Prevalence rates of depression, anxiety disorders, PTSD, bipolar disorder *, schizophrenia *	Representative samples of a general population exposed to conflicts over the previous ten years who resided in their country of origin, were displaced, or had resettled in a neighbouring low-income or middle-income country N = 93.129 subjects included in 70 surveys on depression N = 89,285 subjects studied in 96 surveys on PTSD N = 53,959 subjects evaluated in 38 surveys on anxiety disorders
Hoppen TH, Morina H, 2019 [19]	Meta-Analysis	21/1994–2016	Extrapolation of the absolute number of PTSD and/or MD suffering subjects	N = 7764 war survivors who had lived in an area of conflict up to 25 year prior to the time of the survey
Blackmore R et al., 2020 [20]	Systematic review and meta-analysis	26/2003–2020	Prevalence rates of depression, PTSD, anxiety disorders, schizophrenia	5143 adult refugees and asylum seekers
Ng LC et al., 2020 [21]	Systematic review and meta-analysis	25/2004–2019	Prevalence rates of PTSD	Population representative samples, including refugees (N = 58,887 subjects) living in Sub-Saharan regions
Henkelman JR et al., 2020 [22]	Systematic review and meta-analysis/meta-regression	66/1988–2019	Prevalence rates of diagnosed and self-reported anxiety, depression, PTSD	14,882 adult and child/adolescent refugees in high-income countries
Hoppen TH et al., 2021 [23]	Systematic review and meta-analysis	20/1994–2015	Prevalence rates of PTSD and major depression	Representative samples from countries with a recent history of war (1989/2019): N = 15,420 subjects included in surveys on PTSD, N = 9836 subjects included in surveys on MD and N = 1131 subjects included in surveys on PTSD + MD
Stein J et al., 2021 [24]	Systematic review and meta-analysis	118/2004–2020	Prevalence rates of PTSD	N = 40,188 refugees or internally displaced people from the Eastern Mediterranean Region (EMR) exposed to violent conflict, war and associated human rights abuses
Scoglio AJ and Salhi C, 2021 [25]	Systematic Review	12/2000–2018	Mental health symptoms and/or disorders	N = 4341 resettled refugees exposed to violence
Mesa-Vieira C et al., 2022 [26]	Systematic review and meta-analysis	34/1994–2022	Prevalence rates of common mental disorders	N = 15,549 migrants with pre-migration exposure to armed conflict
Lim ICZY et al., 2022 [27]	Meta-analysis	70/1945–2022	Prevalence rated of Depression, anxiety and PTSD	People exposed to war/violence Studies on depression N = 80,130 Studies on anxiety N = 36,948 Studies of PTSD N = 67,153

PTSD = post-traumatic stress disorder. * Data related to schizophrenia and bipolar disorder are estimated according to the Global Burden of Disease Study 2016.

**Table 2 ijerph-20-02840-t002:** Prevalence rates of mental disorders in systematic reviews/meta-analyses conducted on samples of adult refugees/asylum seekers.

Study	Depression	Anxiety Disorders	Post-Traumatic Stress Disorder	Psychotic Disorders
Fazel M, et al., 2005 [10]	5% *	4% **	9%	2%
Steel Z et al., 2009 [12]	30.8%	N.A. ^	30.6%	N.A. ^
Bogic et al., 2015 [14]	2.3–80.0%	20.3–88.0%	4.4–86.0%	N.A. ^
Blackmore R et al., 2020 [20]	31.5%	11%	31.46%	1.51%
Ng LC et al., 2020 [21]	N.A. ^	N.A. ^	22% ***	N.A. ^
Stein J et al., 2021 [24]	N.A. ^	N.A. ^	31%	N.A. ^
Henkelman JR et al., 2020 [22]	36%	30%	34%	N.A. ^
Scoglio AJ and Salhi C, 2021 [25]	40.2–51.0%	31.8%	20–62% ^^	N.A. ^
Mesa-Vieira C et al., 2022 [26]	25% *	14% **	31%	N.A. ^

The study by Kasmal and Ported [14] reports no prevalence data. * major depression; ** generalized anxiety disorder; N.A. ^ = Not Assessed; *** average prevalence. Prevalence in people exposed to war = 30%; prevalence in those not exposed to war = 8%; ^^ 50% reported PTSD symptoms.

**Table 3 ijerph-20-02840-t003:** Prevalence rates of mental disorders in systematic reviews/meta-analyses conducted on samples of refugees/people living in war/conflict-affected countries.

Study	Depression	Anxiety	Post-Traumatic Stress Disorder	Psychotic Disorder
Ba I and Bhopal RS, 2017 [15]	8.8–76.5%	6.9–75%	3.1–75.9%	N.A. ^
Morina et al. [16]	5.0–80.0%	1.0–81.0%	3.0–88%	1.0–12.0%
Morina N et al., 2018 [17]	27.0%	N.A. ^	26.0%	N.A. ^
Charlson F et al., 2018 [18] *	10.8%	21.7%	15.3%	**
Hoppen TH et al., 2021 [19]	23.1%	N.A. ^	26.5%	N.A. ^
Lim ICZY et al., 2022 [27]	28.9%	30.7%	23.5%	N.A. ^

N.A. ^ = Not Assessed. * The study also included people exposed to war who had resettled in middle- to low-income countries. ** The study does not report average data as only two studies investigating psychosis had been published previously.

**Table 4 ijerph-20-02840-t004:** Systematic reviews and/or meta-analyses relating to the mental health consequences of conflicts on children/adolescents.

Authors, Year of Publication (Reference Number)	Study Type	Number and Time Frame of Papers Considered	Psychopathological Conditions Assessed	Study Samples (N, Type of Populations Considered)
Fazel M, et al., 2005 [10]	Systematic review	5/1996–2002	Prevalence of depression and PTSD	N = 260 refugee adolescents resettled in high-income western countries
Attanayake V et al., 2009 [28]	Systematic review and meta-regression analysis	17/1997–2005	Prevalence rates of PTSD, depression, anxiety disorders	N = 7920 children exposed to war conflicts
Dimitry L, 2012 [29]	Systematic review	71/1989–2010	Prevalence of mental, behavioural and emotional problems in children and adolescents	N = 52,977 children and adolescents exposed to war conflicts in the Middle East
Slone M and Mann S, 2016 [30]	Systematic Review	35/1988–2014	Prevalence of behavioural and emotional symptoms in children	N = 4365 children exposed to war, armed conflicts and terrorism
Kien C et al., 2019 [31]	Systematic review	57/1990–2017	Prevalence of anxiety disorders, depressive disorders, PTSD, emotional and behavioural problems	N = 24,792 asylum-seeking children and adolescents or refugee minors in European countries
Henkelman JR et al., 2020 [22]	Systematic review and meta-analysis	19/1999–2011	Prevalence rates of diagnosed and self-reported anxiety, depression, PTSD	N = 2150 children and adolescent refugees in high-income countries
Blackmore R et al., 2020 [32]	Systematic review and metanalysis	8/2008–2017	Prevalence of PTSD, depression, anxiety disorders, attention-deficit/hyperactivity disorder, oppositional defiant disorder	N = 779 child and adolescent refugees and asylum seekers

**Table 5 ijerph-20-02840-t005:** Prevalence rates of mental disorders in systematic reviews/meta-analyses conducted on samples of children/adolescents.

Study	Depression	Anxiety Dis	PTSD	ADHD	Other
Fazel M et al., 2005 [10]	N.A. ^	N.A. ^	11%	N.A. ^	N.A. ^
Attanayake V et al., 2009 [28]	43%	27%	47%	N.A. ^	N.A. ^
Slone M and Mann S, 2016 [30]	N.A. ^	N.A. ^	37.1% +++	N.A. ^	57.1% @@ 14.4% ## 2.9% §
Dimitry L, 2012 [29]	25–35% * 11.3% ** N.A. ***^	1.4% *°°/1–2 ^^* 40–100 **^^^/28 **” 3.3”, ***/4.3% ***@	5–8% * 23–70% ** 10–30% ***	3% *	1.8% *° 14% **# 2.5% #
Kien C et al., 2019 [31]	20.7%	15%	35.5%	N.A. ^	25.2% + 5.0% ++
Henkelman JR et al., 2020 [22]	28%	32%	52%	N.A. ^	N.A. ^
Blackmore R et al., 2020 [32]	13.8%	15.8%	22.7%	8.6%	1.69% °

N.A. ^ = Not Assessed. * Israel; ** Palestine; *** Iraq. ° oppositional defiant disorder; °° generalized anxiety disorder; ^^ obsessive compulsive disorder; ^^^ unspecified anxiety; “ separation anxiety; # conduct disorder; @ specific phobia; + emotional and behavioural problems (median value); ++ suicidal ideation and behaviours (median value); +++ including PTSD symptoms; @@ behavioural and emotional symptoms.; ## sleep problems; § psychosomatic symptoms.

**Table 6 ijerph-20-02840-t006:** War-related factors impacting mental health.

Exposure to traumatic events and violence (e.g., risks for life, torture, sexual assaults/abuse, witnessing other people killed) Displacement and migration-related stresses (e.g., discrimination, violence socio-economic restraints) Living in refugee camps Loss of loved persons Loss of belongings Loss of work and/or financial stability Food/water shortage

## Data Availability

The present study is a review of published data.

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
