# Peer review of "The Mental Health Costs of Armed Conflicts—A Review of Systematic Reviews Conducted on Refugees, Asylum-Seekers and People Living in War Zones"

_ijerph, 2023, doi:10.3390/ijerph20042840_

Round 1

Reviewer 1 Report

The paper addresses the important topic of the mental health costs for persons living/that have lived in war zones and gives a sound and structured review of systematic reviews / meta-analyses regarding this topic.

The introduction gives a convincing sketch of the relevance of the topic. I only suggest to include the first paragraph of the second chapter in the introduction. This would add the most recent publication to the exposition of the problem and makes the starting point for this review even clearer.

The second chapter “Mental health cost of war for adults” gives a systematic and structured overview of the 22 reviewed systematic reviews / meta-analyses. I suggest to add “and children” to the title of the chapter as the review consists of studies on adults and children. Secondly, I suggest to start the chapter with the first paragraph of the chapter 1.1 where the procedure of building the study corpus for the review is described. In doing so, 1.1 would start with the focus on adults and 1.2. with a focus on children. The systematic review of Ba and Bhopal shows – as others as well – a high range of prevalence rates. It would be helpful if the possible reasons were elaborated in the same way as for other studies. The same would be helpful for the found heterogeneity in Stein et al.

The discussion contextualizes the findings of the review with further recent studies and reflects on the relations of individual health costs with social aspects, especially age and gender. This could be elaborated a bit further as the interconnection of psychic and social aspects especially in the field of forced migration and violence is of utmost importance.

The conclusion is quite short, it would be helpful to add some suggestions for further studies and make any existing limitations visible.

Author Response

REVIEWER 1

The paper addresses the important topic of the mental health costs for persons living/that have lived in war zones and gives a sound and structured review of systematic reviews / meta-analyses regarding this topic.

The introduction gives a convincing sketch of the relevance of the topic. I only suggest to include the first paragraph of the second chapter in the introduction. This would add the most recent publication to the exposition of the problem and makes the starting point for this review even clearer.

The second chapter “Mental health cost of war for adults” gives a systematic and structured overview of the 22 reviewed systematic reviews / meta-analyses. I suggest to add “and children” to the title of the chapter as the review consists of studies on adults and children. Secondly, I suggest to start the chapter with the first paragraph of the chapter 1.1 where the procedure of building the study corpus for the review is described. In doing so, 1.1 would start with the focus on adults and 1.2. with a focus on children

  1. Thanks for these relevant suggestions. The first paragraph of the second chapter was included in the introduction. Moreover, In order to give a better structuration to the paper, after the Introduction we introduced a chapter entitled “Methods”, followed by a chapter entitled “Results”; the latter was divided in a paragraph 1 entitled “Systematic reviews and/or metanalyses relating to adult populations” and in a paragraph 2 entitled “Systematic reviews and/or meta-analyses relating to children/adolescents”. The chapter “Discussion” follows , with  “Conclusions” as a final chapter.

The systematic review of Ba and Bhopal shows – as others as well – a high range of prevalence rates. It would be helpful if the possible reasons were elaborated in the same way as for other studies. The same would be helpful for the found heterogeneity in Stein et al.

  1. As regard the review of Ba and Bophal, following your suggestions, we added the following sentence (lines 187 and following) : “Again, the high variability of reported prevalence rate , according to the same authors, may be due to a series of methodological factors such as the different characteristics of the samples in terms of age, gender, country of origin,  difference in research tools and  the recall period, and to nonmethodological factors, such as the different types of sexual violence suffered by victims, the eventual multiplicity of traumatic experiences” . As regard to Stein et al, we added the sentence (lines 303 and following) “considered the area of the world’s most violent conflicts and the origin of the largest refugee population worldwide.  The pooled estimate of PTS  syndromes was 31% (95% CI: 27 -35 %), with a  considerable heterogeneity and a huge true rate interval  (1%-76%). This heterogeneity in the rates of PTS was attributed by the Authors to a series of factors, analytically evaluated, related to sample characteristics (i.e. sample size, country of origin) , methodological (i.e. decade of study, type of assessment, conflict-related (i.e .stage and severity) and displacement-related factors (present/not present; site of displacement: western or not  western countries)”.

The discussion contextualizes the findings of the review with further recent studies and reflects on the relations of individual health costs with social aspects, especially age and gender. This could be elaborated a bit further as the interconnection of psychic and social aspects especially in the field of forced migration and violence is of utmost importance.

  1. Following your suggestion we expanded the discussion introducing in the texts (lines 467 and following) the sentence “Undoubtely pre-migration traumatic experiences such as those directly linked to war and conflicts are important predictors of negative mental health outcomes but a series of post-migration stressors play a very relevant role for refugees and asylum seekers. Among these socioeconomic factors (i.e. unemployment or underemployment, financial re-strictions/poverty, lack of secure housing), social and interpersonal factors (i.e. family separation, change in previous social role, social isolation, discrimination, loss of social identity. lacking social support, changes in gender role), factor related to the asylum pro-cess and to immigration policies (ie. mandatory detentions, extended processing times, insecure visa status. Lack of access to legal services and representation) have been considered of outmost importance) (37).

The conclusion is quite short, it would be helpful to add some suggestions for further studies and make any existing limitations visible.

  1. This suggestion has given us the possibility to expand significantly the conclusion as follows (lines 518 and following) : “A series of limitations in the current literature should be acknowledged such as the often low quality of studies, the paucity of longitudinal studies, with the consequent difficulty of elucidating the directionality of the relationship between postmigration stressors and mental health consequences , the relative paucity of studies regarding wider ranges of outcomes beyond PTSD, depression and anxiety (32,37). Neverthless, evidences from literature unequivocally demonstrate that war and the consequent displacement are among the most challenging threats to mental health, particularly for the innocent civilians victims of conflicts. Therefore, addressing mental health care for people suffering from war related mental disorders should be considered a priority (52), in particular as regard to children and adolescents (37,53, 54). Unfortunately, the encouraging evidences in favour of the efficacy and acceptability of psychosocial interventions in asylum seekers and refugees are limited to adult populations (55). Moreover, in spite of the availability of effective treatments, on practical grounds the low healthcare access and the poor healthcare ser-vices available  for vulnerable populations such as migrants and refugees (56) are the major obstacles for protecting mental health of these populations. In particular, a series of barriers to mental health care access have been described (33,57,58) such as the lack of knowledge of legal entitlements and the health care system in the hosting country , the poor command of the hosting  country language, the negative cultural beliefs about mental health, the different cultural expectations towards health care professionals, the lack of trust towards services and authorities in the hosting country, and finally the fear of stigma. In order to overcome these barriers and to  improve mental health care a series of recommendations have been suggested (54), namely the ensuring of  education about self-recognition of symptoms of mental health disorders; the overcoming the social and linguistic barriers to access mental health services;  the provision of services for the well-being, augmenting safe, socially acceptable basic services with the help of local support groups and  non-governmental agencies ; the strengthening of the screening capacity for mental health disorders at emergency and primary care level; the provision of specialized services by mental health professionals; the increase of telepsychiatry utilization. Moreover, considering the deleterious impact of post-migration difficulties and the consequent need for psychosocial interventions not limited to a trauma-focused perspective, Li et al (37) suggested policy developments in order to mitigate detrimental post-migration factors through modification of migration and social policies in hosting countries. Examples of these modifications are policies and programs to support refugees to find employment, to build social capital and connections in their new communities, the setting of limitations to prolonged detention in camps and to the ongoing temporary protection. On the same grounds, Evangelidou et al (56), in the context of the MyHealth Project, recommend, at the level of local authorities of destination countries, the facilitation of administrative issues (ie.creation of work or educational opportunities), the promotion of community-based activities of social integration (i.e. using “expert peers”, establishing  health/social services for vulnerable refugees , making available preventive health interventions such as peer support groups, raising awareness in local neighborhood about immigration issues, dis-crimination, racism) and investing in healthcare professional skills (i.e. interpreters, cultural mediators). At level of target populations (migrant/refugees) the actions recommended are those devoted to language acquisition, cultural adaptation of help-seeking behaviours patterns or proactiveness in networking with local communities. Finally, Psychiatrists, and Psychiatric Associations in general are duty-bound to safeguard the mental health of the population (59), not only by taking care of those in whom mental health is impacted by war and post-war-related consequences, but also as a social responsibility to be undertaken through a commitment to raising awareness amongst political decision-makers as to the mental health consequences of armed conflicts and to the mental health policies to be adopted”.

Finally, we are pleased to inform You that the process of revision of the text gave us the opportunity to retrieve a further systematic review published by  Lim ICZY et al (2022), which was included in the tabs.1 and 3 and commented in the “Results” section ( lines 326 and following) ) as follows : “Finally, the meta-analysis of Lim et al (27) report data on prevalence rates of Depressive, Anxiety and Post-traumatic Stress Disorders among people living in war and conflict-affected areas, including military populations and general populations (civilians or refugees). This study is the only one that reports not only the ranges and the aggregate prevalence rates of the disorders considered, but also a series of specific analyses regarding civilian vs military subgroups and studies relating to war periods vs post-war periods. Also this study shows a large variability of prevalence for all disorders, ranging from 3.2 to 79,6% for depression, from 4.2 to 94.8% for anxiety and from 3.9 to 69% for PTSD, but also that the aggregate prevalence rates, according a random-effects model, are higher for depression and anxiety (respectively 28.9% and 30.7%) than for PTSD (23.5%). Interestingly, rates are very significantly higher (p<.001) among civilians than military populations ad regard to depression and anxiety  (respectively 33.% vs 24% and  38.6% vs 16.2% for anxiety; with a non significant difference for PTSD (respectively 25.7% vs 21.3%). Of outmost importance are  the differences of prevalence during war and post-war periods, which are univocally higher (p<001) during war (respectively 38.7% vs 29.1% for Depression, 43.4 vs 30.3% for Anxiety)”.

Moreover, as regard to the same study, in the “Discussion”  we added the sentence (lines 455 and following) “Prevalence of depression and anxiety seems to be even higher than PTSD, and more represented among civilians respect to military personnel; moreover, rates of common mental disorders seems to gradually decrease in post-war periods (27)”

Reviewer 2 Report

This study focuses on the mental health effects of war on adult and child/adolescent refugees and those living in conflict zones, particularly through a systematic review and/or meta-analysis of all published studies from 2005 to the present.

1. Eligibility criteria

Please describe the inclusion and exclusion criteria for the review and how the studies were grouped for integration.

2. Sources of information

List all databases, research registries, websites, organizations, bibliographies, and sources of information researched or consulted to identify the study. List the date each source was last examined.

3. Search Strategy

Describe your complete search strategy for all databases, exam registries, and websites, including any filters or limitations used.

4. Selection process

Describe the methods used to determine whether a given study meets the selection criteria for review, including the number of reviewers who screened each record and each report obtained, whether they worked independently, and if applicable, details of any automated tools used in the process.

5. Data collection process

Describe the methods used to collect data from the reports, including the number of reviewers who collected data from each report, whether they worked independently, the process used to obtain or verify data from the research investigators, and, if applicable, details of any automated tools used in the process.

Author Response

Reviewer 2

Comments and Suggestions for Authors

This study focuses on the mental health effects of war on adult and child/adolescent refugees and those living in conflict zones, particularly through a systematic review and/or meta-analysis of all published studies from 2005 to the present.

  1. Eligibility criteria

Please describe the inclusion and exclusion criteria for the review and how the studies were grouped for integration.

  1. Sources of information

List all databases, research registries, websites, organizations, bibliographies, and sources of information researched or consulted to identify the study. List the date each source was last examined.

  1. Search Strategy

Describe your complete search strategy for all databases, exam registries, and websites, including any filters or limitations used.

  1. Selection process

Describe the methods used to determine whether a given study meets the selection criteria for review, including the number of reviewers who screened each record and each report obtained, whether they worked independently, and if applicable, details of any automated tools used in the process.

  1. Data collection process

Describe the methods used to collect data from the reports, including the number of reviewers who collected data from each report, whether they worked independently, the process used to obtain or verify data from the research investigators, and, if applicable, details of any automated tools used in the process.

  1. Thanks for the opportunity to improve the clarity of the paper, introducing a paragraph regardis “Methods (lines 66-88) ” as follows:

“Methods

  1. Eligibility criteria

 For the purpose of this review, we considered only systematic reviews and/or meta-analyses published in English from 2005 to 2022 reporting quantitative data about prevalence of mental disorders in adult and children living in conflict areas, or refugees/ asylum seekers exposed to war and/or armed conflicts.

  1. Sources of information, search strategy

An electronic search of literature was performed by the Author using Pubmed and Scopus as data bases. The key words used were “armed conflicts” and “mental health”, “armed conflict” and “mental disorders”, “war” and “mental health” “war” and “mental disorders”. 83 and 57 papers were retriewed, respectively, in Scopus and Pubmed,  using  “armed conflicts” and “mental health” as key words; using “armed conflict” and “mental disorders” we retrieved respectively 56 and 51 papers; using “war” and “mental health” we found respectively 50 and 104 papers; and using “war” and “mental disorders” as key words 488 and 114 contributions, respectively, were identified.

  1. Selection and data collection process

Titles and abstract were screened by the Author for inclusion.  Articles rated as possible candidates were added to a preliminary list and their full texts were retriewed and examined . A total of 22 papers were finally selected and considered for this review, following elimination of duplicates and papers not included in the category of systematic reviews and/or meta-analyses.  15 papers are related to adult populations only, 2 papers regard both adults and child/adolescents, 5 of them are related to children and adolescents populations.

Finally, we are pleased to inform You that the process of revision of the text gave us the opportunity to retrieve a further systematic review published by  Lim ICZY et al (2022), which was included in the tabs.1 and 3 and commented in the “Results” section ( lines 326 and following) ) as follows : “Finally, the meta-analysis of Lim et al (27) report data on prevalence rates of Depressive, Anxiety and Post-traumatic Stress Disorders among people living in war and conflict-affected areas, including military populations and general populations (civilians or refugees). This study is the only one that reports not only the ranges and the aggregate prevalence rates of the disorders considered, but also a series of specific analyses regarding civilian vs military subgroups and studies relating to war periods vs post-war periods. Also this study shows a large variability of prevalence for all disorders, ranging from 3.2 to 79,6% for depression, from 4.2 to 94.8% for anxiety and from 3.9 to 69% for PTSD, but also that the aggregate prevalence rates, according a random-effects model, are higher for depression and anxiety (respectively 28.9% and 30.7%) than for PTSD (23.5%). Interestingly, rates are very significantly higher (p<.001) among civilians than military populations ad regard to depression and anxiety  (respectively 33.% vs 24% and  38.6% vs 16.2% for anxiety; with a non significant difference for PTSD (respectively 25.7% vs 21.3%). Of outmost importance are  the differences of prevalence during war and post-war periods, which are univocally higher (p<001) during war (respectively 38.7% vs 29.1% for Depression, 43.4 vs 30.3% for Anxiety)”.

Moreover, as regard to the same study, in the “Discussion”  we added the sentence (lines 455 and following) “Prevalence of depression and anxiety seems to be even higher than PTSD, and more represented among civilians respect to military personnel; moreover, rates of common mental disorders seems to gradually decrease in post-war periods (27)”

Reviewer 3 Report

First of all, I would like to suggest defining “refugees” and “asylum-seeker” statuses, and how the reader could distinguish these categories.

It is worth to also mention that the experience of forcibly displaced youth is varied (Kronick et al., 2017), such as some of them have endured chronic pervasive exposure to interpersonal and community violence, the uncertainty of the future, personal or family persecution, violent loss of loved ones, and an insecure environment, though others have experienced a shorter exposure to high violence such as active war and some youth even come from areas of armed conflict and war with their conscription into the armed forces as “child soldiers”, some arrive without parents or caretakers as unaccompanied and separated minors, and other youths flee with intact families.

There are other important issues I miss from the article:

About one out of three asylum seekers and refugees experience high rates of depression, anxiety, and post-traumatic stress disorder (PTSD) (Turrini et al., 2017).

Early mental health care should therefore be a priority for resettled youth, as post-migration stressors such as prolonged detention, insecure immigration status, and limitations on work and education, can worsen mental health (Li et al., 2016).

When individuals and families seek safety by leaving their homes, cultures, and communities due to the threats of violence and persecution, emotional distress can be heightened (Priebe et al., 2016).

I also suggest that the author consider the recommendations of Chaaya et al. (2022):

Educate about the self-recognition of symptoms of mental health disorders and overcome the social and linguistic barriers to accessing mental health services. Ensure the provision of services for the well-being and augment safe, socially acceptable basic services to protect the dignity of people with the help of local support groups, non-governmental agencies, international coordination, and collaborative mechanisms. Strengthen the capacity to ensure screening for mental health diseases at emergency care visits and primary care health facilities, expand the diversification and training of the mental health workforce across sectors and utilize transcultural approaches to enhance the assessment and provide high-quality health care to the refugees and immigrant populations. Enhance the capacity to provide specialized services by mental health professionals beyond the scope of support groups and primary care providers in an appropriate, accurate timely manner. Utilize the growing telepsychiatry to fill the voids created in health care by the pandemic and war. Remote services play a crucial role in crisis triage and alleviate the pressure on overburdened health systems in war conflicts

Suggested sources and references should be involved in the article:

Bryant, R. A., Schnurr, P. P., & Pedlar, D. (2022). Addressing the mental health needs of civilian combatants in Ukraine. The Lancet Psychiatry9(5), 346–347. https://doi.org/10.1016/S2215-0366(22)00097-9  

Cai, H., Bai, W., Zheng, Y., Zhang, L., Cheung, T., Su, Z., Jackson, T., & Xiang, Y.-T. (2022). International collaboration for addressing mental health crisis among child and adolescent refugees during the Russia-Ukraine war. Asian Journal of Psychiatry72, 103109. https://doi.org/10.1016/j.ajp.2022.103109

Chaaya, C., Devi Thambi, V., Sabuncu, Ö., Abedi, R., Osman Ahmed Osman, A., Uwishema, O., & Onyeaka, H. (2022). Ukraine – Russia crisis and its impacts on the mental health of Ukrainian young people during the COVID-19 pandemic. Annals of Medicine and Surgery79, 104033. https://doi.org/10.1016/j.amsu.2022.104033

Keyes, E. F. Mental health status in refugees: an integrative review of current research. Issues Ment Health Nurs, 21(4), 397–410.

Li, S. S., Liddell, B. J., & Nickerson, A. (2016). The relationship between postmigration stress and psychological disorders in refugees and asylum seekers. Curr Psychiatry Rep, 18(82).

Morris, M. D., Popper, S. T., Rodwell, T. C., Brodine, S. K., & Brouwer, K. C. (2009). Healthcare barriers of refugees post-resettlement. Journal of Community Health34(6), 529–538. https://doi.org/10.1007/s10900-009-9175-3

Priebe, S., Giacco, D., & El-Nagib, R. (2016). WHO Health Evidence Network Synthesis Report 47. Public health aspects of mental health among migrants and refugees: A review of the evidence on mental health care for refugees, asylum seekers and irregular migrants in the WHO European Region. Copenhagen, Denmark: WHO Regional Office for Europe.

Shannon, P. J., Wieling, E., Simmelink-McCleary, J., & Becher, E. (2015). Beyond stigma: Barriers to discussing mental health in refugee populations. Journal of Loss and Trauma20(3), 281–296. https://doi.org/10.1080/15325024.2014.934629

Turrini, G., Purgato, M., Ballette, F., Nose, M., Ostuzzi, G., & Barbui, C. (2017). Common mental disorders in asylum seekers and refugees: umbrella review of prevalence and intervention studies. International Journal of Mental Health Systems, 11(51), 10. 

Author Response

Reviewer 3

Comments and Suggestions for Authors

First of all, I would like to suggest defining “refugees” and “asylum-seeker” statuses, and how the reader could distinguish these categories.

  1. Thanks for giving me the opportunity to be more exahustive. The suggested definitions have been inserted as a note on page 2 : According to Amnesty International Internal displacement is the” involuntary movement of people inside their own country. This movement may be due to a variety of causes, in-cluding natural or human-made disasters, armed conflict, or situations of generalized vio-lence”; the asylum seeker is “an individual who is seeking international protection. In countries with individualised procedures, an asylum seeker is someone whose claim has not yet been finally decided on by the country in which he or she has submitted it. Not every asylum seeker will ultimately be recognised as a refugee, but every refugee is initially an asylum seeker”; a Refugee is “is a person who has fled their country of origin and is unable or unwilling to return because of a well-founded fear of being persecuted because of their race, religion, nationality, membership of a particular social group or political opinion” (https://www.amnesty.org.au/refugee-and-an-asylum-seeker-difference” accessed on 21/1/2023)

It is worth to also mention that the experience of forcibly displaced youth is varied (Kronick et al., 2017), such as some of them have endured chronic pervasive exposure to interpersonal and community violence, the uncertainty of the future, personal or family persecution, violent loss of loved ones, and an insecure environment, though others have experienced a shorter exposure to high violence such as active war and some youth even come from areas of armed conflict and war with their conscription into the armed forces as “child soldiers”, some arrive without parents or caretakers as unaccompanied and separated minors, and other youths flee with intact families.

  1. As regard the points the reviewer underlines, we have inserted in the Discussion, starting from line  489 the following sentence :The severe consequences of armed conflicts for minors emerging from our review should be interpreted the in light of the large variability of individual experiences between different subjects (44)….”

There are other important issues I miss from the article:

About one out of three asylum seekers and refugees experience high rates of depression, anxiety, and post-traumatic stress disorder (PTSD) (Turrini et al., 2017).

  1. As regard to this point , in the Discussion (line 450) we added “…as also shown by the umbrella review from Turrini et al (32) who found point estimates ranging from 4 and 40% for anxiety, from 5 to 44% for depression and from 9 to 36% for PTSD, affecting on the average approximately one third of these populations. Over-all, these data confirm the heavy emotional distress among those who leave their homes, culture and communities due to due to threats of violence and persecution (33)”

Early mental health care should therefore be a priority for resettled youth, as post-migration stressors such as prolonged detention, insecure immigration status, and limitations on work and education, can worsen mental health (Li et al., 2016).

  1. As regard this point, we have included in the Conclusions , a section widely revised also according to reviewer 1 suggestions, we have included (line 525) the following sentence “Therefore, early mental health care for people suffering from war related mental disorders should be considered a priority (52), in particular as regard to children and adolescents (37,53, 54).

When individuals and families seek safety by leaving their homes, cultures, and communities due to the threats of violence and persecution, emotional distress can be heightened (Priebe et al., 2016).

  1. This point was addressed adding in the Discussion (line 453) the following sentence : “Overall, these data confirm the heavy emotional distress among those who leave their homes, culture and communities due to due to threats of violence and persecution (33).

I also suggest that the author consider the recommendations of Chaaya et al. (2022):

Educate about the self-recognition of symptoms of mental health disorders and overcome the social and linguistic barriers to accessing mental health services. Ensure the provision of services for the well-being and augment safe, socially acceptable basic services to protect the dignity of people with the help of local support groups, non-governmental agencies, international coordination, and collaborative mechanisms. Strengthen the capacity to ensure screening for mental health diseases at emergency care visits and primary care health facilities, expand the diversification and training of the mental health workforce across sectors and utilize transcultural approaches to enhance the assessment and provide high-quality health care to the refugees and immigrant populations. Enhance the capacity to provide specialized services by mental health professionals beyond the scope of support groups and primary care providers in an appropriate, accurate timely manner. Utilize the growing telepsychiatry to fill the voids created in health care by the pandemic and war. Remote services play a crucial role in crisis triage and alleviate the pressure on overburdened health systems in war conflicts

  1. R. In the wide revision of Conclusions (see above) we included the following sentence (See lines 538 and following) “In order to overcome these barriers and to improve mental health care a series of recommendations have been suggested (54), namely the ensuring of  education about self-recognition of symptoms of mental health disorders; the overcoming the social and linguistic barriers to access mental health services;  the provision of services for the well-being, augmenting safe, socially acceptable basic services with the help of local support groups and  non-governmental agencies ; the strengthening of the screening capacity for mental health disorders at emergency and primary care level; the provision of specialized services by mental health professionals; the increase of telepsychiatry utilization”.

Suggested sources and references should be involved in the article:

  1. We thanks the reviewer for the many bibliographic suggestions, which allowed us tho improve the overall quality of the paper. We cited the majority of the suggested references, as specified below

Bryant, R. A., Schnurr, P. P., & Pedlar, D. (2022). Addressing the mental health needs of civilian combatants in Ukraine. The Lancet Psychiatry, 9(5), 346–347. https://doi.org/10.1016/S2215-0366(22)00097-9 

  1. See Conclusions (line 526) “Therefore, early mental health care for people suffering from war related mental disorders should be considered a priority (52),

Cai, H., Bai, W., Zheng, Y., Zhang, L., Cheung, T., Su, Z., Jackson, T., & Xiang, Y.-T. (2022). International collaboration for addressing mental health crisis among child and adolescent refugees during the Russia-Ukraine war. Asian Journal of Psychiatry, 72, 103109. https://doi.org/10.1016/j.ajp.2022.103109

  1. See Conclusions (lines 526-7 ): “…. in particular as regard to children and adolescents (, 53, )”.

Chaaya, C., Devi Thambi, V., Sabuncu, Ö., Abedi, R., Osman Ahmed Osman, A., Uwishema, O., & Onyeaka, H. (2022). Ukraine – Russia crisis and its impacts on the mental health of Ukrainian young people during the COVID-19 pandemic. Annals of Medicine and Surgery, 79, 104033. https://doi.org/10.1016/j.amsu.2022.104033

  1. see Conclusions (lines 538 and following) “In order to overcome these barriers and to improve mental health care a series of recommendations have been suggested (54), namely the ensuring of  education about self-recognition of symptoms of mental health disorders; the overcoming the social and linguistic barriers to access mental health services;  the provision of services for the well-being, augmenting safe, socially acceptable basic services with the help of local support groups and  non-governmental agencies ; the strengthening of the screening capacity for mental health disorders at emergency and primary care level; the provision of specialized services by mental health professionals; the increase of telepsychiatry utilization”.

Keyes, E. F. Mental health status in refugees: an integrative review of current research. Issues Ment Health Nurs, 21(4), 397–410.

  1. Not cited

Li, S. S., Liddell, B. J., & Nickerson, A. (2016). The relationship between postmigration stress and psychological disorders in refugees and asylum seekers. Curr Psychiatry Rep, 18(82).

  1. See Concusions (lines 467 and following) : Undoubtely pre-migration traumatic experiences such as those directly linked to war and conflicts are important predictors of negative mental health outcomes but a series of post-migration stressors play a very relevant role for refugees and asylum seekers. Among these socioeconomic factors (i.e. unemployment or underemployment, financial re-strictions/poverty, lack of secure housing), social and interpersonal factors (i.e. family separation, change in previous social role, social isolation, discrimination, loss of social identity, lacking social support, changes in gender role), factors related to the asylum process and to immigration policies (ie. mandatory detentions, extended processing times, insecure visa status, Lack of access to legal services and representation) have been considered of outmost importance) (37).

Morris, M. D., Popper, S. T., Rodwell, T. C., Brodine, S. K., & Brouwer, K. C. (2009). Healthcare barriers of refugees post-resettlement. Journal of Community Health, 34(6), 529–538. https://doi.org/10.1007/s10900-009-9175-3

  1. See Conclusions (lines 532 and following) : “In particular, a series of barriers to mental health care access have been described (…57…) such as the lack of knowledge of legal entitlements and the health care system in the hosting country , the poor command of the hosting country language, the negative cultural beliefs about mental health, the different cultural expectations towards health care professionals, the lack of trust towards services and authorities in the hosting country, and finally the fear of stigma

Priebe, S., Giacco, D., & El-Nagib, R. (2016). WHO Health Evidence Network Synthesis Report 47. Public health aspects of mental health among migrants and refugees: A review of the evidence on mental health care for refugees, asylum seekers and irregular migrants in the WHO European Region. Copenhagen, Denmark: WHO Regional Office for Europe.

  1. See Discussion (lines 453 and following) : “Overall, these data confirm the heavy emotional distress among those who leave their homes, culture and communities due to due to threats of violence and persecution (33)” and Conclusions (lines 532 and following) “In particular, a series of barriers to mental health care access have been described… (33…)”

Shannon, P. J., Wieling, E., Simmelink-McCleary, J., & Becher, E. (2015). Beyond stigma: Barriers to discussing mental health in refugee populations. Journal of Loss and Trauma, 20(3), 281–296. https://doi.org/10.1080/15325024.2014.934629

R  See Conclusions (lines 532 and following) : “In particular, a series of barriers to mental health care access have been described… (….58)

 Turrini, G., Purgato, M., Ballette, F., Nose, M., Ostuzzi, G., & Barbui, C. (2017). Common mental disorders in asylum seekers and refugees: umbrella review of prevalence and intervention studies. International Journal of Mental Health Systems, 11(51), 10. 

  1. see the Discussion (line 450 and following ) “…as also shown by the umbrella review from Turrini et al (32) who found point estimates ranging from 4 and 40% for anxiety, from 5 to 44% for depression and from 9 to 36% for PTSD, affecting on the average approximately one third of these populations”

Take note that we cited also Turrini G, Purgato M, Acarturk M et al, Efficacy and acceptability of psyhosocial interventions in asylum seekers and refugees: systematic re-view and menta-analysies, Epidemiology Psychiatric Sciences, 2019, 28:376-388, https: // doi.org/10.1017/S2045796019000027 (ref 55)

Finally, we are pleased  to inform You that the process of revision of the text gave us the opportunity to retrieve a further systematic review published by  Lim ICZY et al (2022), which was included in the tabs.1 and 3 and commented in the “Results” section ( lines 326 and following) ) as follows : “Finally, the meta-analysis of Lim et al (27) report data on prevalence rates of Depressive, Anxiety and Post-traumatic Stress Disorders among people living in war and conflict-affected areas, including military populations and general populations (civilians or refugees). This study is the only one that reports not only the ranges and the aggregate prevalence rates of the disorders considered, but also a series of specific analyses regarding civilian vs military subgroups and studies relating to war periods vs post-war periods. Also this study shows a large variability of prevalence for all disorders, ranging from 3.2 to 79,6% for depression, from 4.2 to 94.8% for anxiety and from 3.9 to 69% for PTSD, but also that the aggregate prevalence rates, according a random-effects model, are higher for depression and anxiety (respectively 28.9% and 30.7%) than for PTSD (23.5%). Interestingly, rates are very significantly higher (p<.001) among civilians than military populations ad regard to depression and anxiety  (respectively 33.% vs 24% and  38.6% vs 16.2% for anxiety; with a non significant difference for PTSD (respectively 25.7% vs 21.3%). Of outmost importance are  the differences of prevalence during war and post-war periods, which are univocally higher (p<001) during war (respectively 38.7% vs 29.1% for Depression, 43.4 vs 30.3% for Anxiety)”.

Moreover, as regard to the same study, in the “Discussion”  we added the sentence (lines 455 and following) “Prevalence of depression and anxiety seems to be even higher than PTSD, and more represented among civilians respect to military personnel; moreover, rates of common mental disorders seems to gradually decrease in post-war periods (27)”